# Knowledge and awareness of bovine tuberculosis associated with raw milk and under-cooked meat contamination among cattle farmers in selected parts of zambia

**Anthony Phiri**[1]*, **Emmanuel Likulunga**[2], **Adriace Chauwa**[3], **Mildred Zulu**[4],
**Beatrice Kankhuni**[5], **Ngula Monde**[6], **Sydney Malama**[2]

**1** Department of Disease Control, School of Veterinary Medicine, University of Zambia, Lusaka, Zambia,
**2** Department of Biological Sciences, School of Natural Sciences, University of Zambia, Lusaka, Zambia,
**3** Enteric Disease and Vaccine Research Unit, Centre for Infectious Disease Research in Zambia, Lusaka,
Zambia, **4** Department of Pathology and Microbiology, School of Medicine, University of Zambia, Lusaka,
Zambia, **5** Department of Clinical Medicine, Levy Mwanawasa Medical University, Lusaka, Zambia,
**6** Department of Biomedical Sciences, Tropical Disease Research Centre, Ndola, Zambia

* anthonyphiri2002@gmail.com

## ABSTRACT

### Background

Bovine tuberculosis (bTB) is a disease of cattle that is transmitted through direct contact with an infected animal or ingestion of contaminated food.

### Methods

A mixed-methods study was conducted in two districts of Zambia, Lundazi and Monze, from December 2021 to June 2022. A cross-sectional survey with 280 respondents, five focus group discussions, and five key informant interviews were conducted. Descriptive statistics were used to analyze quantitative data using R software, while qualitative data was analyzed using Nvivo and content analysis.

### Results

Social and cultural practices were reported to significantly contribute to bTB transmission with risky behaviors including the consumption of undercooked meat and unpasteurized milk, and inadequate protective measures during cattle slaughtering. Overall, 75.3% of male and 70.3% of female respondents expressed a poor level of awareness of bTB transmission. Among cattle farmers, 85.7% lacked formal education, resulting in low knowledge levels, with 99.5% expressing poor awareness. Additionally, findings revealed that 99.5% of cattle farmers had poor awareness of bTB, human doctors/clinicians, and veterinary/para-veterinarians had 100% excellent knowledge. Abattoir workers had good knowledge at 78.3% and cattle businessmen had a fair knowledge level at 96.4%. Overall, bTB awareness varied across occupations, with human and animal healthcare professionals being the most knowledgeable.

**Data availability statement:** Dataset generated during the current study are available in the supplementary materials section of this manuscript

**Funding:** The author(s) received no specific funding for this work.

**Competing interests:** The authors have declared that no competing interests exist.

## Conclusions

Our study found low levels of awareness and misperception about bTB among cattle farmers. Awareness varied across professionals with educated individuals being more knowledgeable. The consumption of undercooked meat and raw milk highlighted these knowledge gaps. Therefore, it is crucial to conduct public health campaigns to raise awareness about bTB causes, sources of infection, and control measures.

## Author Summary

Bovine tuberculosis (bTB) is a significant public health concern in Zambia, particularly among cattle farmers who handle and consume raw milk and undercooked meat. The authors conducted a cross-sectional survey among cattle farmers, abattoir workers, cattle businessmen, human doctors, and veterinarians. Quantitative data found that cattle farmers demonstrated poor awareness of bTB transmission, with 75.3% of males and 70.3% of females expressing inadequate knowledge. A significant proportion (85.7%) of cattle farmers lacked formal education, contributing to low knowledge levels. Human doctors and veterinarians exhibited excellent knowledge (100%) of bTB, while abattoir workers demonstrated good knowledge (78.3%) and cattle businessmen showed fair knowledge (96.4%). Qualitative data found that people in the study areas consume unpasteurized milk. It was further, revealed that they handle meat with bare hands, sell uninspected meat, and consume undercooked meat.

Therefore, the researchers, suggested that there is a need to launch public awareness campaigns through various media channels to educate the general public about bTB risks and prevention measures. Integrate bTB awareness into the curriculum of local schools to educate young people about the disease, and strengthen the enforcement of meat inspection regulations to ensure that all meat sold in markets is inspected and certified by veterinary authorities.

## Introduction

Tuberculosis (TB) is an infectious disease that affects both humans and certain animal species, particularly those used for dairy and meat products, such as cattle, sheep, and goats [1]. The disease occurs most commonly in developing countries, where the consumption of unpasteurized dairy products, occupational exposure to infected livestock, and the absence of regular bovine TB (bTB) testing programs in livestock systems are prevalent [2]. Additionally, the risk of bTB transmission is higher in endemic areas where people, such as farmers, veterinarians, and abattoir workers are more likely to have direct contact with animals [3].

   Globally, Tuberculosis significantly impacts human health, resulting in over 3.5 million deaths annually with bTB being the cause of 3% of these cases [4]. In Zambia, the involvement of *Mycobacterium bovis* (*M. bovis)* in the national TB burden is not yet known. Additionally, the situation is complicated by inadequate or nonexistent institutional support systems and a lack of control and research facilities. The responsibility for controlling the transmission of diseases that are not considered "diseases of national economic importance" (DNEIs), despite their severe public health impacts, falls entirely in the hands of cattle owners. In Africa, it is estimated that about 7% of all human TB cases are ascribed to *M. bovis* (Anaelom, 2010).

Additionally, 26.0% and 57.0% of the disease burden are disproportionately in Africa and Asia, respectively [4].

Zambia produces an estimated 253 million liters of milk yearly, with only 44 million liters processed through formal processing channels. An estimated 209 million liters are consumed raw, with more than half of this milk coming from the Southern province [5]. Further statistics indicate that an estimated 80% of Zambia's 3 million cattle are owned by traditional farmers, who rely on these animals for milk, which is usually unpasteurized[5.] Additionally, the consumption of unpasteurized milk remains a major practice in the rural communities. Hence this, behavioral practice poses a health risk if the milk is drawn from infected animals [6]

This interface tends to create a hot spot for the spread of bTB [7]. Consequently, bTB has been reported to be an endemic challenge in Zambian traditional cattle farming, with a high herd prevalence of 49.8% recorded in areas within, and adjacent to, the Kafue basin as far back as 1947 [8]. Findings from abattoirs in the Namwala districts indicated that 16.8% of slaughtered cattle were infected with bTB as evidenced by the presence of typical TB lesions [9].

Additionally, the consumption of raw milk [10] is a major common practice in the reported communities. However, Pandey [11] showed that the prevalence of bTB in the Southern province was reduced compared to the studies conducted by Munyeme [12] (2009), who reported a 6.8%, prevalence rate whilst Pandey reported an estimated prevalence of 2.6% [11,12].

Therefore, there is a need to design control programs to achieve the World Health Organization's 'END-TB' agenda, aiming to eliminate human TB as a public health concern by 2035 (WHO,2021).[10] However, One Health disease mitigation strategy can be supported by the Health Belief Model's (HBM) "Perceived Vulnerability" construct, which states that individuals are more likely to adopt healthier behaviors when they feel personally at risk or vulnerable. The stronger the risk perception of an illness, the more likely people are to act and participate in behavioral change that would reduce the disease's risk [13].

The objective of this study was to assess awareness levels of bovine tuberculosis among cattle farmers regarding undercooked meat, and raw meat consumption. The study aimed to categorize bTB awareness levels as "Excellent", "Good", "Fair", and "Poor" and to identify transmission factors such as under-cooked meat consumption, well-cooked meat consumption, unpasteurized milk consumption, pasteurized milk consumption, handling of beef with bare hands, and consuming uninspected beef. The outcomes of this study are expected to contribute valuable information to the surveillance and control efforts aligned with the WHO's 'END-TB' plan, which plans to eliminate all forms of human TB by 2035 [10]

The study also significantly contributes to the management practices of bovine tuberculosis (bTB) in line with the main motivation in the fields of animal and human health, which is to enhance the understanding of the effectiveness of the current government regulatory framework on bTB in Zambia. Further, the study findings fill a knowledge gap in the level of awareness of bTB among cattle farmers and veterinary professionals in Zambia. Previous studies have focused on the prevalence of bTB in Zambia, however, very few have examined awareness levels of bTB among stakeholders

### Null and alternative hypotheses

Null Hypothesis(H0): There is no significant difference in the level of awareness about the risks associated with bTB transmission through raw milk and undercooked meat contamination between cattle farmers and veterinarians

Alternative Hypothesis (H1): Cattle farmers will have a lower level of awareness about the risks associated with bTB transmission through raw milk and undercooked meat contamination compared to veterinarians

## Method

### Ethics statement

The research proposal was approved by the University of Zambia's Biomedical Research Ethics Committee. Ethical clearance number 2102–2021. Permission to conduct a study was granted by the respective village Headmen. The participants were told about the nature of the study and their rights, which included the right to personal privacy, the right to withdraw from the research at any time without providing a reason, and the right to review and withhold interview content, all following the principles of informed consent and voluntary participation. Furthermore, there was no physical or emotional damage to the participants. The study was founded on the principles of secrecy, anonymity, and confidentiality.

### Study design and site

The study was conducted in Lundazi and Monze districts of Zambia. A total of twenty villages were purposively selected. The nine villages were selected from Lundazi (Engeleweni, Kacindila, Efumbeni, Gowa, Themba, Katoto, Chizungu, Chipetuka, and Bokosi) and eleven villages were selected from Monze (Munababa, Mazunkha, Shimbayi, Mwanakaba, Hamusonde, Shimbwaalale, Mwando, Chavwa, Hankogote, Hashote, and Chilomo). The elected villages were located close to veterinarian camps, which allowed the researcher to access other relevant information about the disease, additionally, villages were accessible by roads, which made it easier for the research team to conduct the study, that in Lundazi the studies were conducted during the rainy season. Hence some roads were impassable

The proposed study sites were chosen due to their large cattle populations and high levels of human-animal interaction [14] making them appropriate areas for this research. Lundazi district is located in the Eastern province of Zambia, which had a total cattle population of 597,149 with Lundazi district contributing 63,144 of this total number. Monze district is located in the Southern province of Zambia which has a total cattle population of 1,225,090, and Monze district contributes 284,713 to this number [14]. Additionally, these are the provinces where residents tend to consume a lot of unpasteurized milk [15].

### Sampling method

A purposive sampling method was used to select respondents within the villages. The researcher selected livestock farmers who own cattle and are directly involved in the managing and handling of raw milk and undercooked meat.

### Sample Size

### Data collection

All interviews lasted close to 60 minutes, which was well within the recommended 90 minutes for an interview [16] Two methods were used to record the data. To begin, the researcher acted as both a recorder and an interviewer by keeping track of the responses in a field notebook. Second, with the permission and consent of the interviewees, the researcher used a voice recorder to record the interviews. Data gathering was a big challenge, due to time and resource constraints, the study could not cover all the cattle farmers in Zambia but focused mainly on cattle farmers in the selected areas of Eastern and Southern provinces. Not all targeted villages were captured in the Eastern province because during the researcher's visit it was rainy season, many roads were impassable and many cattle farmers were busy tilling their land

**Quantitative.** A total number of 208 households were selected for this study. Household heads were interviewed using a structured questionnaire with mostly close-ended questions.

Each household was represented by one respondent, therefore only one was interviewed per household. The study included both males and females aged 18 years and above. In addition, 72 key informants who were included were categorized as follows: 11 veterinary/ para-veterinarians, 10 human doctors/clinicians, 23 abattoir workers, and 28 commercial businessmen. In total, 280 respondents were recruited of which 119 were from Eastern province and 161 from Southern province. The questionnaire was translated into local languages, *Chichewa* for Lundazi district and *Chitonga* for Monze district. Pre-designed questionnaires were pre-tested from Chisamba district. Before collecting data, veterinary officers were oriented to the questionnaire by the Principal Investigator to ensure consistency in data collection. Data was collected through face-to-face interviews. Questions asked involved whether respondents had ever heard of bovine tuberculosis, awareness of the mode of transmission, and who gets infected.

**Qualitative.** To gather data and understand the circumstances in which the disease occurs and is transmitted, qualitative data was collected. as follows:

Five (5) focus group discussions (FGDs) were held as follows: In Lundazi (n = 5), and in Monze (n = 5). Focus group participants were purposefully sampled from among survey respondents. Focus group discussions were managed using interview guides with open-ended questions to allow respondents to comment on responses in their own words. Further, questions asked concerned what respondents knew about bTB transmission from animal to human, how an individual knows that a family member is infected with TB, signs and symptoms of the disease in humans and animals, mode of transmission of bTB, preventive measures and challenges that they face in controlling the disease, meat consumption practices, and milk consumption practice. All discussions were held in Chichewa and Chitonga as appropriate. Consequently, the responses were recorded on a digital recorder after getting verbal consent from the participants.

## Analytical methods

**Quantitative.** Descriptive statistics on Socio-demographic characteristics, awareness levels, and factors of transmission towards bovine tuberculosis on undercooked meat and raw milk consumption among cattle farmers were performed and analyzed in R software [17] with the "ordinal" R package [18]. Multinomial logistic regression was used to determine the relationship between various factors including meat consumption, milk consumption, and meat handling.

**Qualitative.** All Audio files that were in focus group discussions were later translated into English language and thereafter transcribed into computer files. After a repeat reading of the narratives, the data was later transferred into Nvivo for coding. At first, broad coding was managed using major themes that were extracted from the topic guides. Subsequently, it was followed by another coding from the major themes into sub-themes. Framework matrices were then put together by identifying key quotes under each sub-theme and summarizing the narratives under each major theme. Framework matrices were used to cross-check narratives from the key informants with those of the focus group participants to identify divergent or supporting views. Illustrative quotations that represented the themes were used in the data results.

## Results

### Quantitative

**Socio-demographic characteristics.** Two hundred and eighty (280) participants were selected to determine knowledge awareness. Of these, 13.21% were female and 86.79% were male (Fig 1). Study results revealed more males than females in cattle farming because cattle

farming and operations often require physically demanding work. In addition, in Zambian cattle farming, men are expected to take on more responsibilities in livestock farming, while women are directly involved in household management.

This study indicated that a significant proportion of individuals across various categories except those with a medical or veterinary specialty exhibited poor knowledge of bTB awareness (p < 0.001, OR = 3.12, CI = 1.96–4.97) Table 1.

The study revealed no veterinary personnel were observed to consume meat (either well or unwell-cooked). Table 2

**Qualitative.** Four main themes were yielded from the focus group discussions and interviews as below;

- Knowledge about bTB transmission

- Unpasteurized milk and uncooked meat consumption

- Handling beef with bare hands

- Attitudes of cattle farmers toward consumption of unpasteurized milk and undercooked meat consumption

**Knowledge about bTB transmission.** Cattle farmers showed 99.5% poor knowledge about bTB and its transmission from animal to human. However, they associate bTB with what is called chifuba chachikulu in Lundazi (strong cough) they would only associate and assume the disease to clinical signs like weight loss, weakness, and fever. The study findings revealed

Key: Dark brown color represents Male Respondents and light brown represents Female respondents

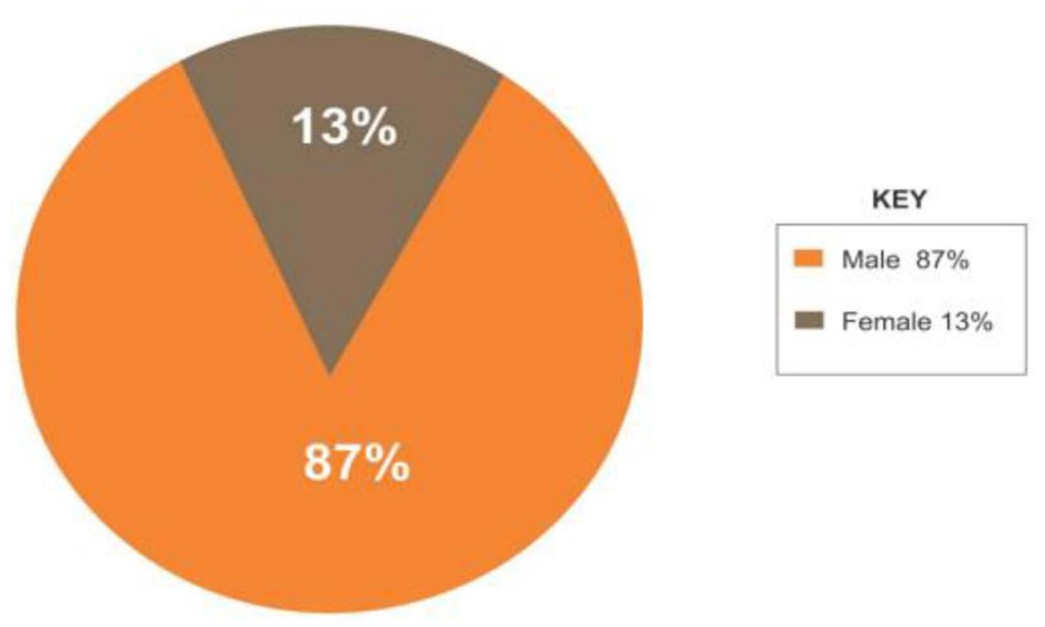

**Fig 1. Showing the percentage of Males and Females in the study areas.** Authors' field data.

**Table 1. Summary of the distribution of social demographic characteristics of the study population based on bovine tuberculosis awareness levels (Excellent, Good, Fair, and Poor).**

| Variable | Category | N | Excellent (%) | Good (%) | Fair (%) | Poor (%) |
|---|---|---|---|---|---|---|
| Gender | Male | 243 | 19 (7.8) | 13 (5.3) | 28 (11.5) | 183 (75.3) |
| | Female | 37 | 2 (5.4) | 5 (13.5) | 4 (10.8) | 26 (70.3) |
| Age | <30 | 59 | 2 (3.4) | 4 (6.8) | 5 (8.5) | 48 (81.4) |
| | 30–40 | 50 | 7 (14.0) | 7 (14.0) | 8 (16.0) | 28 (56.0) |
| | 41–50 | 35 | 10 (28.6) | 5 (14.3) | 4 (11.4) | 16 (45.7) |
| | >50 | 136 | 2 (1.5) | 2 (1.5) | 15 (11.0) | 117 (86.0) |
| Education | No education | 28 | 0 (0.0) | 0 (0.0) | 4 (14.3) | 24 (85.7) |
| | Primary | 40 | 1 (2.5) | 0 (0.0) | 1 (2.5) | 38 (95.0) |
| | Secondary | 87 | 10 (11.5) | 18 (20.7) | 14 (16.1) | 45 (51.7) |
| | College | 99 | 43 (43.0) | 13 (13.0) | 30 (30.0) | 14 (14.0) |
| | University | 21 | 14 (66.7) | 5 (23.8.) | 5 (23.8) | 2 (9.5) |
| | Postgraduate | 5 | 5 (100) | 0 (0.0) | 0 (0.0) | 0 (0.0) |
| Occupation | Human doctors/clinicians | 10 | 10 (100) | 0 (0.0) | 0 (0.0) | 0 (0.0) |
| | Veterinary/para-veterinarians | 11 | 11 (100) | 0 (0.0) | 0 (0.0) | 0 (0.0) |
| | Abattoir workers | 23 | 0 (0.0) | 18 (78.3) | 4 (17.4) | 1 (4.3) |
| | Commercial business people | 28 | 0 (0.0) | 0 (0.0) | 27 (96.4) | 1 (3.6) |
| | Cattle farmers | 208 | 0 (0.0) | 0 (0.0) | 1 (0.5) | 207 (99.5) |

N: shows the total number of participants in each category with the proportions (%) provided in brackets.

**Table 2. Summary of the distribution of social demographic characteristics of the study population based on bovine tuberculosis factors that may affect transmission.**

| Variable | Category | N | UMC (%) | CMC (%) | UMIC (%) | PMIC (%) | HBBH (%) | UBC (%) |
|---|---|---|---|---|---|---|---|---|
| Gender | Male | 243 | 48((19.8) | 20 (8.2) | 66 (27.2) | 16 (6.6) | 40 (16.5) | 53(21.8) |
| | Female | 37 | 14(37.8) | 3 (8.1) | 9 (24.3) | 2 (5.4) | 7 (18.9) | 2(5.4) |
| Age | <30 | 59 | 17 (28.8) | 4 (6.8) | 16 (27.1) | 2 (3.4) | 10 (16.9) | 10 (16.9) |
| | 30–40 | 50 | 8 (16.0) | 6 (12.0) | 10 (20.0) | 7 (14.0) | 9 (18.0) | 10 (20.0) |
| | 41–50 | 35 | 4 (11.4) | **5 (14.3)** | 13 (37.1) | 4 (11.4) | 0(0.00) | 9 (25.7) |
| | >50 | 136 | 33 (24.3) | 13 (9.6) | 36 (26.5) | 5(3.7) | 23 (16.9) | 26 (19.1) |
| Education | No education | 28 | 5 (17.9) | 5 (17.9) | 9 (32.1) | 2 (7.1) | 4 (14.3) | 3 (10.7) |
| | Primary | 40 | 12 (30.0) | 4 (10.0) | 11 (27.5) | 3 (7.5) | 4 (10.0) | 6 (15.0) |
| | Secondary | 87 | 12 (13.8) | 6 (6.9) | 25 (28.7) | 11 (12.6) | 16 (18.4) | 17 (19.5) |
| | College | 99 | 30 (30.3) | 7 (7.1) | 24 (24.2) | 2 (2.0) | 13 (13.1) | 23 (23.2) |
| | University | 21 | 2 (9.5) | **10 (47.6)** | 0 (0.00) | 5(23.8) | 0(0.00) | 4 (19.0) |
| | Postgraduate | 5 | 1 (20.0) | 1 (20.0) | 1 (20.0) | **0 (0.0)** | **0 (0.0)** | 2 (40.0) |
| Occupation | Human doctors/clinicians | 10 | 1 (10.0) | 3 (30.0) | 1(10.1) | **0 (0.0)** | 1 (10.0) | 4 (40.0) |
| | Veterinary/para-veterinarians | 11 | **0 (0.0)** | **4 (36.4)** | 0 (0.0) | 7 (63.6) | **0 (0.0)** | 0 (0.0) |
| | Abattoir workers | 23 | 7 (30.4) | 1 (4.3) | 1 (43) | 6 (26.1) | 5 (21.7) | 3 (13.0) |
| | Commercial business people | 28 | **16 (57.2)** | **0 (0.0)** | 0 (0.00) | 5 (17.9) | 7 (25.0) | 0(0.00) |
| | Cattle farmers | 208 | 60 (28.8) | 21 (10.1) | 53 (25.5) | 5 (2.4) | 34 (16.3) | 35 (16.8) |

N: total number of participants in each category, UMC: under-cooked meat consumption, CMC: well-cooked meat consumption, UMIC: unpasteurized milk consumption, PMIC: pasteurized milk consumption, HBBH: handling of beef with bare hands, UBC: Uninspected beef consumption

that the respondents understood common signs for bTB in Lundazi and Monze unfortunately could not know how it is transmitted.

> *"All along, I could not tell that my animal was suffering from bovine tuberculosis, I only knew when I took my dead animal to the abattoir. They conducted a post-mortem inspection and characteristic tuberculous lesions were found in the lungs, liver, and other parts. Initially, I used to slaughter animals and consume most of the meat without making consideration of the lesions."*
>
> *(Male participant; Lundazi)*

Respondents were able to identify clinical signs.

> *"An animal with a strong cough would lose weight, the animal would be weak, lose appetite, and have a fluctuating fever."*
>
> *(Female participant; Monze)*
>
> *"Animal loss of weight and lack of appetite are all associated with a strong cough……. No one can get any disease by consuming unpasteurized milk and undercooked meat consumption from cattle suffering a strong cough."*
>
> *(Female participant; Monze)*

No one among the respondents was able to discuss the transmission of bovine tuberculosis from animal to human. Generally, awareness of bovine tuberculosis was very poor among cattle farmers.

**Unpasteurized milk and uncooked meat consumption.** Qualitative findings also revealed that people in the two study areas, Lundazi and Monze commonly consume unpasteurized milk. Additionally, it was found that they sold uninspected meat and consumed undercooked meat.

> *"When sending children to school, they would give them breakfast inform of unpasteurized milk they would also slaughter and sell their animals without being inspected by health personnel and they would consume meat from taverns that had not been fully cooked."*
>
> *(Veterinary officer; Monze)*
>
> *Unpasteurized milk and undercooked meat are significant risk factors for bTB transmission. There is a need to advise cattle farmers and the general public about these risks."*
>
> *(Veterinarian officer; Lundazi)*
>
> *"Bovine tuberculosis can be transmitted from animals to humans. I have observed that cattle farmers have been consuming raw milk and undercooked meat, therefore we need to take this seriously and educate the farmers and public about the risks."*
>
> *(Humam doctor; Monze)*

Many of the respondents would consume unpasteurized milk and undercooked meat,

> *"I grew up feeding on unpasteurized milk and feeding on meat that was not fully cooked. I would consume meat that has not been fully prepared from beer-drinking places."*
>
> *(Female participant; Monze)*
>
> *"When an animal has been unwell, we always slaughter and consume the meat. Some of the meat will be roasted whilst some meat will be fully cooked."*

*(Male participant; Lundazi)*

**Handling beef with bare hands.** Some of the significant responses from the respondent that they handled meat with bare hands as this had been their custom ever since time immemorial.

> *"When I slaughter my animals, I don't use anything to protect my hands since I believe that the internal parts of my animals are clean and free from diseases. So why should I put on something to protect my hands?"*

*(Male respondent; Monze)*

> *"I do not wear gloves or protective material when handling meat. I believe the meat is clean and I'm clean too. As long as I wash my hands it's all fine."*

*(Commercial businessman; Lundazi)*

> *"We need better training on how to handle meat safely and prevent the spread of diseases like bTB."*

*(Male abattoir worker, Monze)*

### Attitudes of cattle farmers toward consumption of unpasteurized milk and undercooked meat consumption

Veterinarian officers revealed that most cattle farmers had a positive attitude toward the consumption of unpasteurized milk and consumption of undercooked meat. Veterinarians/ para-veterinarians indicated that some of the challenges they faced in bTB eradication and control were failure to persuade cattle farmers against the consumption of unpasteurized milk and raw meat consumption as stipulated below:

> *Cattle farmers would say……."It is our culture to consume unpasteurized milk. My great-grandparents have been consuming unpasteurized milk and never got sick. It is the more reason, that I shall continue consuming unpasteurized milk. And no one would come to stop me. My family is not sick though we eat unpasteurized milk on a daily basis, in fact, boiling milk destroys the quality of the milk".*

*(Veterinary officer, Monze)*

> *"Consumption of meat in Zambia is a very strong cultural behavior. Meat is always consumed as part of the staple diet of the people as well as during special occasions of festivity. It's related to cultural symbolic weight that is greater than any other food. Consumption of raw meat or half-cooked meat, Lubende/ Chilopa (consumption of blood) is very common in Lundazi."*

*(Veterinary officer; Lundazi)*

### Discussion

This study set out to investigate knowledge awareness of the transmission of bTB among cattle farmers in Lundazi and Monze districts of Zambia. Findings from these two districts were very similar and hence could not conduct a comparative analysis. The current study reports that social and cultural practices were reported as major risky practices for bTB transmission. These factors included the consumption of undercooked meat, the consumption of raw milk, and lack of protective measures during the slaughtering of cattle.

A survey conducted by the Zambia Demographic and Health Survey [19] revealed that the proportion of males and females is almost the same. However, our findings have shown male respondents to be the majority in the study. This could be a result of the fact that females are less likely to be listed as household heads but also that males dominate the role of herding cattle. [14]

Our demographic factors state that respondents below 30 years were found to have 81.4% poor awareness levels. This might be because most of the youthful age did not have formal education (85.7%) and this might have contributed to their lack of awareness. Youthful respondents in the study area may prioritize very immediate needs, such as clothes, food, and shelter, over long-term investments in bTB prevention and control they may have limited exposure to cattle farming and bovine tuberculous. However, this finding contrasts with Yapei [20] who argued that youths are generally more educated, gather information from social media, and form a network that enhances their health knowledge. Our findings also showed 86% poor awareness levels of respondents in the age of 50 years and above. This could be a result of a lack of access to media and low education levels, further, may not have had regular interactions with veterinary professionals, which could limit their opportunities to learn about bTB. Elderly respondents may have experienced memory decline, which could affect their ability to learn and retain new information [19] this is, however, contrary to the findings of Yusul [21] who reported that older cattle farmers are educated, very much aware of animal diseases, have vast experience and better practices regarding zoonotic disease prevention and management in comparison to younger ones.

Further, our study results showed that awareness levels differed across educational levels, with those without education reported to be 85.7% and those respondents with primary education reported to be at 95% poor awareness levels of bTB. These findings are similar to a study conducted in Ethiopia by Adugna [22] who stated that the main reasons for poor awareness of bovine tuberculosis in cattle farmers could be as a result of low education levels, weak health institution linkages, and inadequate mass media lessons. Findings also revealed that low education levels may be more likely to live in absolute poverty, which can limit access to information about bovine tuberculosis. They may also hold traditional beliefs and practices that do not prioritize bTB prevention and control. Similar findings were also reported by Amare et al., [23] in Ethiopia who stated that livestock farmers had a poor level of understanding of bTB. This was a result of low education levels and poor programs concerning zoonotic diseases such as tuberculosis, brucellosis, anthrax, and food safety. However, a study by Kuma et al., [24] in South West Ethiopia indicated a higher level of knowledge about bTB among livestock farmers without formal education and those with primary school education. This may be due to respondents living close to major cities, where they have access to the latest information from various media sources such as radio, television, and educational campaigns from healthcare institutions [25]. Additionally, Munyeme et al. [26], found that the lack of awareness among cattle farmers was due to insufficient information about transmission of the diseases between animals and humans.

Similarly, qualitative study findings also indicated low bTB awareness levels which differ across occupations. The results revealed low literacy levels among livestock farmers. It revealed that human doctors/clinicians, veterinary/para-veterinarians, and postgraduate students were highly knowledgeable about bTB compared to other occupational categories. This is similar to the study by Lindahl [27] in Tajikistan, Senegal, Nepal, and India who reported that cattle farmers had a low level of awareness, attitude, and practices toward zoonotic diseases in comparison to those professionals with high education levels.

In agreement, our quantitative data reported that 99.5% of cattle farmers had poor levels of bTB awareness levels. Further, abattoir workers expressed a good knowledge level at 78.3%,

Our qualitative results also stipulate that at abattoirs, workers with some education in meat hygiene handling, were knowledgeable about the transmission of zoonotic diseases hence put on Personal Protective Equipment (PPEs), whilst others were not protecting themselves. These findings support those of Ghali [28], who showed that some abattoir workers had mixed perceptions, about zoonotic disease. This inconsistency explains why some workers wore gloves and other protective materials to mitigate the risk of zoonotic disease transmission while others did not. Furthermore, cattle businessmen expressed fair knowledge levels at 96.4%. The study findings are also in agreement with Ghali et al. [28] who stated that many Butchers are not knowledgeable about the risk of selling meat and consuming meat that is not inspected. However, veterinary/para veterinarians and human doctors/ clinicians had 100% excellent awareness levels. Our study findings were similar to those documented in Ethiopia [23] and Adugna [22] which showed a higher proportion of knowledge among medical technicians about bTB transmission and infection similarly, recent study findings by Sadiq, [29] who assessed the knowledge and opinion of cattle farmers in Malaysia indicated that those with higher education and in medical fields had a better understanding of zoonoses.

The current study also revealed that 28.8% of cattle farmers consume undercooked meat and 25.5% consume unpasteurized milk, whilst 0.0% of veterinary/para-veterinarians consume undercooked meat and unpasteurized meat respectively. A further 10% and 10.1% of human doctors/clinicians consume undercooked and unpasteurized milk. This finding is similar to that reported by Biru [30] who found that medical technicians had low consumption levels of under-cooked meat as well as raw milk because of the kind of job in which they give service to the animals almost daily and hence have no appetite for meat consumption.

This study's findings revealed that no significant differences were observed in the bTB transmission factors in gender, and age. However, factors of transmission were observed in education as well as in the occupation. Respondents with high education levels recorded a low transmission mode compared to those with informal education. This study's findings are similar to the study conducted by Cassidy et al. [31] who showed that consumption of under-cooked meat and unpasteurized milk was common in his study population due to low literacy levels and high poverty levels. Similarly, qualitative findings revealed that cattle farmers due to low literacy levels consume raw milk. In addition, they handled meat and meat products without protective materials, consistent with study findings by Youssef et al. [32] who showed that several cattle farmers at an abattoir tested positive for bTB infection after being examined by tuberculin intradermal test and ELISA.

The current study findings about occupational categories showed that consumption of uncooked meat and raw milk was extremely rare among veterinary/para-veterinarians, who reported 0.00% for both undercooked meat and unpasteurized milk consumption. However, 10% of human doctors and clinicians reported consuming undercooked meat and unpasteurized meat. The absence of consumption among veterinary/para-veterinary professionals can be attributed to their high level of knowledge about the disease and the nature of their work [23]. In contrast, qualitative data indicated that the minimal consumption of meat and milk by human doctors/ clinicians often occurs at events or parties where they might consume roasted meat or unpasteurized milk as breakfast. Additionally, 57.2% of cattle businessmen reported consuming undercooked meat, which aligns with the WHO [4] report that Nigerian meat sellers tend to eat meat raw meat in front of their customers to demonstrate that the meat on display is safe for consumption.

## Conclusion

The study findings revealed that social and cultural practices were reported as major risky practices for bTB transmission to people, such as the consumption of undercooked meat, consumption of unpasteurized milk, and lack of protective measures during the slaughtering

of cattle. Further, awareness towards bTB varied across education and professional with some educated ones being more aware of the disease than those not educated. Consequently, those professionals related to health were very much aware and had a perfect attitude toward the disease.

To address these awareness levels and attitudes towards bovine tuberculosis in raw milk and under-cooked meat among cattle farmers in Zambia, the research suggests the following recommendations:

  i.  Develop and implement targeted educational campaigns aimed at cattle farmers to improve their knowledge and awareness of bTB transmission, prevention, and management. These campaigns should utilize local languages (Chichewa for the Eastern province and Chitonga for the Southern province) and focus on the importance of boiling milk and cooking meat thoroughly.

 ii.  Utilize local radio stations, community meetings, and agricultural extension services to disseminate information about bTB. This approach will ensure that information reaches a wider audience, including those who do not have access to formal education or traditional media.

iii.  Integrate bTB awareness into the curriculum of local schools to educate young people about the disease, its transmission, and prevention. This will help in creating long-term awareness within the community.

 iv.  Provide ongoing training for veterinarians, para-veterinarians, and human health clinicians on the latest methods of diagnosing, managing, and preventing bTB. This will ensure that these key informants are well-equipped to educate and support cattle farmers.

  v.  Organize workshops and seminars for abattoir workers, cattle businessmen, and other stakeholders involved in the cattle industry to enhance their understanding of bTB and safe meat handling practices.

 vi.  Enhance diagnostic facilities in rural areas to ensure timely and accurate detection of bTB in cattle. This can involve mobile veterinary clinics and improved laboratory services.

vii.  Strengthen the enforcement of meat inspection regulations to ensure that all meat sold in markets is inspected and certified by veterinary authorities. This will reduce the consumption of potentially infected meat.

## Limitations

The findings from a sample size of 208 may not be representative of the larger population, which can limit the generalization of the finding, women are very much involved in household management, therefore cultural limitations may have prevented women from participating in the study, which can limit the understanding of their perspectives and experiences. Further, the study was only conducted in two districts, which may not be representative of the entire nation. The study may have been subject to measurement and systematic errors, such self self-reporting, social desirability bias, or recall bias. However, the study attempted to ensure internal validity and reliability by triangulation of quantitative and qualitative findings. Lastly, the study was not designed to estimate the prevalence of bovine tuberculosis, therefore, further study to conduct a nationwide study to determine the prevalence and incidence of bovine tuberculosis in Zambia should be undertaken.

## Supporting information

**S1 Dataset.** **This dataset presents information on the knowledge and awareness of bovine tuberculosis (bTB) association with raw milk and under-cooked meat contamination among cattle farmers in selected parts of Zambia.**
(XLSX)

## Author contributions

**Conceptualization:** Adriace Chauwa, Sydney Malama.

**Data curation:** Anthony Phiri, Mildred Zulu, Sydney Malama.

**Investigation:** Anthony Phiri.

**Methodology:** Emmanuel Likulunga.

**Project administration:** Anthony Phiri.

**Software:** Emmanuel Likulunga.

**Validation:** Ngula Monde.

**Writing – original draft:** Anthony Phiri.

**Writing – review & editing:** Adriace Chauwa, Mildred Zulu, Beatrice Kankhuni, Ngula Monde.

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
