## [Decision Letter · Decision Letter 0]

19 Feb 2025

Knowledge and Awareness of Bovine Tuberculosis Associated with Raw Milk and Under-Cooked Meat Contamination Among Cattle Farmers in Selected Parts of Zambia.

Dear Dr. Phiri,

Thank you for submitting your manuscript to PLOS Neglected Tropical Diseases. After careful consideration, we feel that it has merit but does not fully meet PLOS Neglected Tropical Diseases's publication criteria as it currently stands. Therefore, we invite you to submit a revised version of the manuscript that addresses the points raised during the review process.

Please submit your revised manuscript within 60 days Apr 20 2025 11:59PM. If you will need more time than this to complete your revisions, please reply to this message or contact the journal office at plosntds@plos.org. Please include the following items when submitting your revised manuscript:

We look forward to receiving your revised manuscript.

Kind regards,

Joseph M. Vinetz

Section Editor

Joseph Vinetz

Section Editor

Shaden Kamhawi

co-Editor-in-Chief

Paul Brindley

co-Editor-in-Chief

**Journal Requirements:**

- ® on page: 1.

5) Tables should not be uploaded as individual files. Please remove these files and include the Tables in your manuscript file as editable, cell-based objects. For more information about how to format tables, see our guidelines:

https://journals.plos.org/plosntds/s/tables

6) We have noticed that you have uploaded Supporting Information files, but you have not included a list of legends. Please add a full list of legends for your Supporting Information files after the references list.

7) Some material included in your submission may be copyrighted. According to PLOSu2019s copyright policy, authors who use figures or other material (e.g., graphics, clipart, maps) from another author or copyright holder must demonstrate or obtain permission to publish this material under the Creative Commons Attribution 4.0 International (CC BY 4.0) License used by PLOS journals. Please closely review the details of PLOSu2019s copyright requirements here: PLOS Licenses and Copyright. If you need to request permissions from a copyright holder, you may use PLOS's Copyright Content Permission form.

Potential Copyright Issues:

- Figure 1. Please provide a direct link to the base layer of the map (i.e., the country or region border shape) and ensure this is also included in the figure legend; and provide a link to the terms of use / license information for the base layer image or shapefile. We cannot publish proprietary or copyrighted maps (e.g. Google Maps, Mapquest) and the terms of use for your map base layer must be compatible with our CC BY 4.0 license.

8) We note that your Data Availability Statement is currently as follows: "N/A". Please confirm at this time whether or not your submission contains all raw data required to replicate the results of your study. Authors must share the “minimal data set” for their submission. PLOS defines the minimal data set to consist of the data required to replicate all study findings reported in the article, as well as related metadata and methods (https://journals.plos.org/plosone/s/data-availability#loc-minimal-data-set-definition).

- The points extracted from images for analysis..

**Reviewers' Comments:**

Reviewer's Responses to Questions

**Key Review Criteria Required for Acceptance?**

**Methods** :

-Are the objectives of the study clearly articulated with a clear testable hypothesis stated?

-Is the study design appropriate to address the stated objectives?

-Is the population clearly described and appropriate for the hypothesis being tested?

-Is the sample size sufficient to ensure adequate power to address the hypothesis being tested?

-Were correct statistical analysis used to support conclusions?

-Are there concerns about ethical or regulatory requirements being met?

Reviewer #1: 1. The objectives of the study are somewhat implied, but there isn’t a clearly stated, testable hypothesis in the typical format. The aim is to assess the level of awareness regarding bovine tuberculosis and identify the factors contributing to its transmission, but a more specific, testable hypothesis would strengthen the research design. For example: "Cattle farmers have lower awareness of bTB transmission compared to veterinary professionals."

2. Yes, the mixed-methods design (quantitative through surveys and qualitative through focus groups) is appropriate for addressing the study’s objectives. The quantitative approach allows for measuring awareness levels across different groups, while the qualitative approach provides in-depth insights into the cultural and social practices that may contribute to bTB transmission.

3. Yes, the population is clearly described. The study includes respondents from two districts in Zambia (Lundazi and Monze) with cattle farmers, abattoir workers, veterinary professionals, and commercial businessmen. The focus on these groups is appropriate for understanding the awareness of bTB transmission and the factors contributing to its spread, as they are directly involved with cattle handling and consumption of animal products.

4. The sample size appears to be adequate for a descriptive cross-sectional study. A total of 280 respondents, including both quantitative survey participants and qualitative key informants, is a reasonable number for drawing conclusions on awareness levels. However, the power of the study would depend on the statistical analysis and the size of the subgroups being compared (e.g., cattle farmers vs. veterinary professionals). The sample size for the qualitative aspects (focus groups and key informants) is smaller, but this is typical for qualitative data.

5. Yes, appropriate statistical methods were used. Descriptive statistics were used to summarize the data, and multinomial logistic regression was applied to assess the relationships between factors such as meat consumption, milk consumption, and handling. The use of R software for analysis is standard and appropriate for handling the quantitative data. For qualitative data, Nvivo was used for coding and identifying themes, which is a valid method for qualitative analysis.

6. Yes, ethical considerations were appropriately addressed. The study received approval from the University of Zambia's Biomedical Research Ethics Committee, and participants were informed about the study's nature and their rights, including the right to privacy, voluntary participation, and informed consent. These ethical measures seem to be in line with common standards for human research.

**Results** :

-Does the analysis presented match the analysis plan?

-Are the results clearly and completely presented?

-Are the figures (Tables, Images) of sufficient quality for clarity?

Reviewer #1: 1. Yes, the analysis presented matches the analysis plan outlined in the methodology. Quantitative data was analyzed using descriptive statistics, and multinomial logistic regression, which is consistent with the described approach. The qualitative data was analyzed by coding the focus group discussions and interviews into major themes and sub-themes using Nvivo, which aligns with the planned qualitative analysis approach. The results from both quantitative and qualitative analyses are clearly presented in the study.

2. Yes, the results are presented clearly, but there could be some improvements in terms of completeness and detail. For example:

The quantitative results are summarized in terms of statistical significance (p-value < 0.001) and the odds ratio (OR = 3.12), which provide an understanding of the difference in knowledge between various groups (such as cattle farmers versus veterinary professionals).

The qualitative findings are organized by themes (knowledge of bTB, unpasteurized milk and uncooked meat consumption, and handling beef with bare hands). Specific quotes from participants are included to illustrate these themes, which strengthens the depth of the findings. However, a more detailed presentation of the results, such as how the statistical tests (e.g., multinomial logistic regression) were used to assess the relationship between knowledge and behaviors, would enhance the clarity of the results. Including specific data tables (like Table 1) and providing the exact significance levels or confidence intervals would offer more comprehensive insight.

3. Yes, Figures and Tables provide sufficient detail for the reader to understand the results easily.

**Conclusions** :

-Are the conclusions supported by the data presented?

-Are the limitations of analysis clearly described?

-Do the authors discuss how these data can be helpful to advance our understanding of the topic under study?

-Is public health relevance addressed?

Reviewer #1: 1. Yes, the conclusions are supported by the data presented in the study. The findings regarding low awareness of bTB and the risky behaviors (such as consumption of undercooked meat and unpasteurized milk) are directly linked to the conclusions. The study highlighted the importance of education and professional background in awareness of bTB, which is reflected in the recommendation to target educational campaigns at cattle farmers and integrate bTB awareness into local schools. The suggestions for improving diagnostic facilities and strengthening enforcement of regulations are based on the identified gaps in knowledge and practices related to bTB prevention.

2. The limitations of the analysis are not explicitly described in the Conclusion section. However, in the Methods and Results, some limitations could be inferred (e.g., reliance on self-reporting, regional differences in awareness, and sample size), but they are not directly addressed in the conclusion. For a more robust conclusion, the authors could have mentioned potential limitations, such as the cross-sectional nature of the study, limited sample size, or challenges in generalizing the findings to the broader population.

3. Yes, the authors discuss how the data can help advance understanding by emphasizing the need for educational campaigns to improve awareness and behavior regarding bTB. By identifying the gaps in knowledge, the authors suggest practical interventions (e.g., local radio campaigns, school curricula) to increase understanding of bTB transmission and prevention. These recommendations are aimed at reducing risky behaviors and improving health practices among cattle farmers and the broader community, thereby advancing our understanding of how to address bTB effectively in rural Zambia.

4.Yes, public health relevance is well-addressed in the conclusion. The study emphasizes the importance of improving awareness about bTB transmission, prevention, and management, particularly in high-risk populations like cattle farmers. The recommendations for educational campaigns, improved veterinary services, and stricter meat inspection are directly aimed at improving public health outcomes by reducing the risk of bTB transmission. These interventions would not only benefit cattle farmers but also help prevent zoonotic transmission to the wider population, thus contributing to public health improvements.

**Editorial and Data Presentation Modifications?**

Reviewer #1: Considerations:

ACCEPTED WITH MINOR CORRECTION

Reason to accept:

1. The study provides valuable insight into cattle farmers awareness of the transmission of bTB, focusing on cultural and social practice that pose risks that highly relevant to public health and agricultural sectors, especially in developing country.

2. The article contributes to a deeper understanding of how demographic, educational, and occupational factors affect knowledge about bTb.

**Summary and General Comments** :

Reviewer #1: The following issue should be revised by the Author to make the manuscript publishable. The comments are as follow:

Abstract:

Overall, the abstract provides a clear overview of the study’s key finding, with practical implications for public health in Zambia.

Introduction:

The background on TB and bTB is informative but could benefit from a more focused introduction to the study’s specific objectives. Explicitly mention how the study contributes to existing research knowledge gaps in Zambia.

Methodology:

1. The rationale for choosing Lundazi and Monze districts is well-explained, highlighting the relevance of high cattle population and human-animal interaction. However, consider providing more context on the selection of villages within the districts, for example were they also based on any specific disease burden or other criteria?

2. The methodology on data collection is thorough. A minor suggestion would be to clarify how the respondents were selected. Was it purely random sampling within villages, or is there any stratification occur (e. g., by occupation or exposure risk)?

3. More information on the practical aspects of data collection, such as the duration of interviews or any challenges faced during fieldwork, would add depth to the methodology section.

Results:

The Quantitative results: The sociodemographic data is clear. It would be helpful to present a brief explanation of why a significant proportion of males (86.79%) participated, as this could indicate gender-related patterns in this study area.

The Qualitative results: The four key themes are well-defined, and the qualitative data offers valuable insight into local knowledge and practices. However, the richness of qualitative data could be better conveyed if more illustrative quotes were presented, especially to show variations across different subgroup (e.g., farmers vs veterinarians)

Discussion and conclusion:

1. in some places, there are contradiction between your findings and those from other studies. For instance, the comparison between your study and Yusul’s findings regarding age and education levels could benefit from further elaboration on why such discrepancies might exist.

2. There is no mention of potential limitations or gaps in the data, such as the absence of data from woman (due to cultural reasons or the study design) or the fact that your study was conducted only in two districts. Moreover, limitations on the sample size or methodological constraints should be acknowledged in the discussion.

Final comments:

Overall, the study provides valuable insight into bTB awareness in Zambia and offers important recommendations for improving knowledge and prevention. The findings are important for understanding zoonotic disease transmission, and the practical recommendations have the potential to make real impact.

PLOS authors have the option to publish the peer review history of their article (what does this mean? ). If published, this will include your full peer review and any attached files.

**Do you want your identity to be public for this peer review?** For information about this choice, including consent withdrawal, please see our Privacy Policy .

Reviewer #1: No

**Figure resubmission:**

**Reproducibility:**



---

## [Editor Report · Decision Letter 1]

13 Mar 2025

Response to Reviewers
Revised Manuscript with Track Changes
Manuscript

Shaden Kamhawi

co-Editor-in-Chief

Paul Brindley

co-Editor-in-Chief

**Journal Requirements:**

1) Please upload all main figures as separate Figure files in .tif or .eps format. For more information about how to convert and format your figure files please see our guidelines: 

2) We have noticed that you have uploaded Supporting Information files, but you have not included a list of legends. Please add a full list of legends for your Supporting Information files after the references list.

**Reviewers' comments:****Figure resubmission:****Reproducibility:** To enhance the reproducibility of your results, we recommend that authors of applicable studies deposit laboratory protocols in protocols.io, where a protocol can be assigned its own identifier (DOI) such that it can be cited independently in the future. Additionally, PLOS ONE offers an option to publish peer-reviewed clinical study protocols. Read more information on sharing protocols at https://plos.org/protocols?utm_medium=editorial-email&utm_source=authorletters&utm_campaign=protocols

---

## [Editor Report · Decision Letter 2]

20 Mar 2025

Dear Phiri,

We are pleased to inform you that your manuscript 'Knowledge and Awareness of Bovine Tuberculosis Associated with Raw Milk and Under-Cooked Meat Contamination Among Cattle Farmers in Selected Parts of Zambia.' has been provisionally accepted for publication in PLOS Neglected Tropical Diseases.

Best regards,

Joseph M. Vinetz

Section Editor

Joseph Vinetz

Section Editor

Shaden Kamhawi

co-Editor-in-Chief

Paul Brindley

co-Editor-in-Chief

---

## [Editor Report · Acceptance letter]

Dear Phiri,

We are delighted to inform you that your manuscript, "Knowledge and Awareness of Bovine Tuberculosis Associated with Raw Milk and Under-Cooked Meat Contamination Among Cattle Farmers in Selected Parts of Zambia.," has been formally accepted for publication in PLOS Neglected Tropical Diseases.

Best regards,

Shaden Kamhawi

co-Editor-in-Chief

Paul Brindley

co-Editor-in-Chief
